# How Relevant Are Bone Marrow-Derived Mast Cells (BMMCs) as Models for Tissue Mast Cells? A Comparative Transcriptome Analysis of BMMCs and Peritoneal Mast Cells

**DOI:** 10.3390/cells9092118

**Published:** 2020-09-17

**Authors:** Srinivas Akula, Aida Paivandy, Zhirong Fu, Michael Thorpe, Gunnar Pejler, Lars Hellman

**Affiliations:** 1Department of Cell and Molecular Biology, Uppsala University, The Biomedical Center, Box 596, SE-751 24 Uppsala, Sweden; srinivas.akula@icm.uu.se (S.A.); fuzhirong.zju@gmail.com (Z.F.); getmeinahalfpipe@gmail.com (M.T.); 2Department of Medical Biochemistry and Microbiology, Uppsala University, The Biomedical Center, Box 589, SE-751 23 Uppsala, Sweden; aida.paivandy@imbim.uu.se (A.P.); gunnar.pejler@imbim.uu.se (G.P.); 3Department of Anatomy, Physiology and Biochemistry, Swedish University of Agricultural Sciences, Box 7011, SE-75007 Uppsala, Sweden

**Keywords:** in vitro model, transcriptome, mRNA, mast cell, tryptase, chymase, serine protease, FcεRI, heparin, histamine

## Abstract

Bone marrow-derived mast cells (BMMCs) are often used as a model system for studies of the role of MCs in health and disease. These cells are relatively easy to obtain from total bone marrow cells by culturing under the influence of IL-3 or stem cell factor (SCF). After 3 to 4 weeks in culture, a nearly homogenous cell population of toluidine blue-positive cells are often obtained. However, the question is how relevant equivalents these cells are to normal tissue MCs. By comparing the total transcriptome of purified peritoneal MCs with BMMCs, here we obtained a comparative view of these cells. We found several important transcripts that were expressed at very high levels in peritoneal MCs, but were almost totally absent from the BMMCs, including the major chymotryptic granule protease Mcpt4, the neurotrophin receptor Gfra2, the substance P receptor Mrgprb2, the metalloprotease Adamts9 and the complement factor 2 (C2). In addition, there were a number of other molecules that were expressed at much higher levels in peritoneal MCs than in BMMCs, including the transcription factors Myb and Meis2, the MilR1 (Allergin), Hdc (Histidine decarboxylase), Tarm1 and the IL-3 receptor alpha chain. We also found many transcripts that were highly expressed in BMMCs but were absent or expressed at low levels in the peritoneal MCs. However, there were also numerous MC-related transcripts that were expressed at similar levels in the two populations of cells, but almost absent in peritoneal macrophages and B cells. These results reveal that the transcriptome of BMMCs shows many similarities, but also many differences to that of tissue MCs. BMMCs can thereby serve as suitable models in many settings concerning the biology of MCs, but our findings also emphasize that great care should be taken when extrapolating findings from BMMCs to the in vivo function of tissue-resident MCs.

## 1. Introduction

Mast cells (MCs) are tissue-resident cells that are known primarily for their prominent role in IgE-mediated allergies [1,2]. They are evolutionary old and MCs or MC-like cells are present in all studied vertebrates. MC-like cells have also been identified in tunicates, a chordate, which is an early ancestor to the vertebrates as represented by the sea squirt *Ciona intestialis* [3]. These cells have been shown to stain positive with toluidine blue, similar to mammalian MCs, and to contain heparin, histamine and are capable of producing prostaglandin D2 as well as expressing a serine protease with tryptic activity [4]. It is also interesting to note that no humans with a complete lack of MCs have been identified, indicating an important physiological role for these cells. MCs are often positioned at the interphase between the tissue and environment, where they can act as part of our first line of defense. There they can trigger inflammation and attract other inflammatory cells to the area of inflammation. In both rodents and humans, two major subtypes of MCs are found. One of them is primarily found in connective tissues and such cells are therefore named connective tissue MCs (CTMCs). High numbers of CTMCs are found in skin and the tongue, and lower numbers in many other organs [5]. The second type, which differs in several aspects concerning granule content and surface receptors, is found at mucosal sites, such as the intestinal mucosa and in humans also in the lungs, and are therefore named mucosal MCs (MMCs) [6,7,8,9,10,11]. Mature MCs are generally only found in tissues and not in the circulation.

MCs of both the connective tissue and the mucosal type store large numbers of cytoplasmic granules that are rapidly exocytosed following activation. These granules contain massive amounts of proteases, primarily serine proteases, but in CTMCs also a MC-specific carboxypeptidase, named carboxypeptidase A3 (CPA3). The serine proteases expressed by MCs can generally be subdivided into chymases and tryptases [12,13,14,15]. Chymases are chymotrypsin-like and cleave substrates on the C-terminal side of aromatic amino acids whereas tryptases are trypsin-like in their specificity, with a preference for Arg and Lys in the P1 position [12,13,14,15]. Very high amounts of these proteases are found in MCs, where the levels can reach 35% of the total cellular protein [16]. These proteases have been shown to inactivate snake, bee and scorpion toxins, regulate blood pressure by angiotensin II generation, and to control inflammation by cleaving a selective panel of cytokines [1,14,17,18,19,20,21]. Mast cell proteases likely have several other important physiological functions, including connective tissue turnover and regulating coagulation [22,23].

MC granules also contain high levels of heavily sulfated and thereby negatively charged, glycosaminoglycans such as heparin or chondroitin sulfate, and also large amounts of vasoactive, low-molecular weight compounds including histamine and serotonin [16,24,25,26,27]. A number of cell surface receptors and other molecules, of which many are MC-specific or restricted to a few cell types are also expressed by MCs. Important such receptors are the high affinity receptor for IgE (FcεRI) and the receptors for stem cell factor (SCF) and IL-3 [28,29]. More recently, MCs have also been found to express a receptor for low-molecular weight and positively charged substances like substance P, the Mrgprb2 receptor in mice and its human counterpart MRGPRX2 [30]. MCs in different tissues, such as CTMCs and MMCs, show major differences in their expression of granule proteases, levels of receptors and in numerous other aspects, indicating that they have, at least partly, different physiological functions.

For most cell types there are great difficulties to obtain relevant in vitro models. However, MCs have been regarded as one exception due to the possibility to generate cells that by many criteria appear to resemble tissue MCs, by growing bone marrow cells in the presence of IL-3 or SCF [31]. After several weeks of culturing, most other cell types die off and the remaining cells are to almost 100% MC-like cells, so called bone marrow-derived MCs (BMMCs). They stain positive for classical mast cell-staining dyes like Alcian blue and Toluidine blue and express the high affinity receptor for IgE [31,32,33,34]. However, an important question is how similar these cells are to tissue-resident MCs. Recently we performed a large transcriptomal analysis of MCs at different tissue locations with the focus on peritoneal MCs, but also included an analysis of BMMCs grown in the presence of IL-3 (5). We showed that BMMCs represented relatively immature cells of the MC lineage, with low levels of several MC-specific proteases, including the major CTMC chymase, Mcpt4, and the major CTMC tryptase, Mcpt6 (5). However, differences between peritoneal MCs and BMMCs were not extensively studied. Therefore, to obtain a more multi-faceted view of the phenotypic differences between purified mouse peritoneal MCs and the in vitro-differentiated BMMCs, here we present an extended analysis of this issue. Our analysis identifies a number of important differences in the transcriptome of these two MC populations, and also identifies numerous transcripts that are restricted to MCs in comparison to macrophages and B cells. This information can serve as a basis for judgements of the relevance of using BMMCs as a model to study MC function.

## 2. Materials and Methods

### 2.1. Mice

Female BALB/c mice were purchased from Taconic Biosciences (Ejby, Denmark) and maintained at the animal facility in the Biomedical Center (Uppsala University) or the National Veterinary Institute (Uppsala, Sweden). The animal experiments were approved by the local ethical committee (Uppsala djurförsöksetiska nämnd; Dnr 5.8.18-05357/2018).

### 2.2. Generation of Bone Marrow-Derived MCs and LPS Stimulation

Bone marrow cells were isolated from the femur and tibia of mice and grown in Dulbecco’s modified Eagle’s medium (Sigma-Aldrich, Saint Louis, MO, USA) containing 30% WEHI-3B-conditoned medium, 10% heat-inactivated fetal bovine serum (BSA, Gibco, Carlsbad, CA, USA), 100 U/mL penicillin, 100 μg/mL streptomycin, 2 mM L-glutamine (all from Sigma-Aldrich) and 10 ng/mL recombinant IL-3 (PeproTech, Rocky Hill, NJ, USA). The medium was changed twice every week and cells were cultured at a concentration of 0.5 × 10^6^ cells/mL in a humidified 37 °C incubator with 5% CO_2_ for at least four weeks to obtain mature and pure BMMCs. The cells were divided into two separate fractions, one was directly frozen in liquid nitrogen for preparation of total RNA and the second was incubated in the above medium with the addition of 1 μg of LPS/mL for 4 h (Sigma-Aldrich L4516—from *E. coli* O127:B8), after which the cells were pelleted and frozen in liquid nitrogen for subsequent RNA preparation.

### 2.3. Peritoneal Cell Extraction and MACS Sorting of Peritoneal Mast Cells

For extraction of peritoneal cells, naïve mice were euthanized by isoflurane overdose and neck dislocation, the abdominal skin was removed and 9 mL ice-cold phosphate-buffered saline (PBS) was injected into the peritoneal cavity. After ensuring that the injected PBS is thoroughly dispersed within the peritoneal cavity, peritoneal lavage fluid was collected and the cells were pelleted by centrifugation at 400× *g* for 10 min. The cells were pooled and resuspended in magnetic-activated cell sorting (MACS) buffer, containing 0.5% BSA in PBS (pH 7.2) and 2 mM EDTA, followed by incubation with 20 μL c-kit (CD117) MicroBeads (Miltenyi Biotec, Bergish Gladbach, Germany). After 30 min, cells were washed, resuspended in MACS buffer and passed through a LS column (Miltenyi Biotec). Magnetically labeled (c-kit^+^) cells were collected and used for RNA isolation and assessment of MC purity. Alternatively, to enhance the purity of peritoneal MCs, one step of negative selection was performed before c-kit positive selection. Briefly, collected peritoneal cells were incubated in the first step with 1 μL of primary PE-Cy5-conjugated antibodies against lineage surface markers including CD3 (17A2), CD4 (GK1.5), CD8b (eBioH35-17.2), CD11b (M1/70), CD19 (ebio1D3), B220 (RA3-6B2), Gr-1 (RB6-8C5), and TER-119 (TER-119) from BD Biosciences (Franklin Lakes, NJ, USA) or eBioscience (Hatfield, United Kingdom). After 20 min, the cells were washed, resuspended in MACS buffer and incubated with 20 μL anti-Cy5/anti-Alexa Fluor 647 MicroBeads (Miltenyi Biotec) for 30 min. After washing, the cells were resuspended in MACS buffer and passed through a LD column (Miltenyi Biotec) according to the manufacturer’s instructions. Subsequently, the unlabeled (Lin^−^) cells were collected, washed and in the second step incubated with 20 μL c-kit (CD117) MicroBeads for 30 min. After washing, cells were resuspended in MACS buffer and passed through a LS column (Miltenyi Biotec). The unlabeled cells were discarded while magnetically labeled (Lin^−^ c-kit^+^) cells were collected and used for RNA isolation and assessment of MC purity.

### 2.4. FACS Sorting of Peritoneal Macrophages and B Cells

Peritoneal cells were obtained and prepared as described above and resuspended in PBS (pH 7.4) with 2% heat-inactivated fetal bovine serum (Gibco), followed by incubation with the fluorochrome-labelled anti-CD19 (1D3), anti-CD11b (M1/70) and anti-F4/80 (BM8), obtained from BD Biosciences or eBioscience. The flow cytometry-based sorting of peritoneal macrophages and B cells was performed on a FACSAria III (BD Biosciences) and data were analyzed with FlowJo software (TreeStar Inc., Ashland, OR, USA).

### 2.5. Image Analysis

The magnetically sorted Lin^−^ CD117^+^ cells were cytospun onto glass slides using a Shandon Cytospin 2 (Thermo Fisher Scientific, Inc., Waltham, MA, USA) and were allowed to dry before staining with toluidine blue using a standard protocol to assess the purity of sorted Lin^−^ CD117^+^ MCs. The cells were imaged using an Eclipse Ni-U microscope (Nikon, Tokyo, Japan, 100× magnification).

### 2.6. RNA Isolation from Purified Cell Fractions

Total RNA was prepared from MACS- and FACS-sorted cells as well as cultured untreated and LPS-treated BMMCs using the Nucleospin RNA kit (Macherey-Nagel, Düren, Germany), according to the manufacturer’s recommendations. The RNA was eluted with 30 μL of DEPC-treated water, the RNA concentration was determined using a Nanodrop ND-1000 (Nano Drop Technologies, Wilmington, Delaware, DE, USA). The integrity of the RNA was confirmed by visualization on 1.2% agarose gels using ethidium bromide staining.

### 2.7. Analysis of the Transcriptome by the Thermo Fisher Ampliseq PCR Based Method

To determine the transcriptome of the different cell fractions we used the Thermo Fisher chip-based Ampliseq transcriptomic platform (Ion-Torrent next-generation sequencing system—Thermofisher.com). This analysis platform is based on the purification on a chip of the individual mRNAs (as cDNAs), which are then PCR-amplified and sequenced individually. Before binding to the chip, the RNA is not fragmented but copied into cDNAs and every mRNA is read only once. The number of reads therefore corresponds to the expression level.

## 3. Results

### 3.1. Preparation of RNA from Purified Peritoneal Cell Fractions

To study the difference in total transcriptome of in vitro-differentiated and mature tissue MCs, we prepared relatively highly purified populations of both of these cells. In order to compare expression levels with other hematopoietic cells we also purified macrophages and B cells. One source of MCs, macrophages and B cells in mice is the peritoneal cavity. In the peritoneum, macrophages and B cells are the dominating cell types, constituting approximately 30–40% and 40% of the entire peritoneal cell populations, respectively [35]. Here, MCs are less abundant and represent ~1–2% of the peritoneal cells. The MCs found in the peritoneum are almost identical in their phenotype to classical skin MCs concerning their major granule protease content, and both of these MC populations are classified as CTMCs.

MCs were purified by both negative and positive selection involving several steps of MACS, resulting in a purity of over 95% as well as sufficient amounts of cells to obtain a good coverage of the entire transcriptome (Figure 1A). Peritoneal macrophages (primarily large peritoneal macrophages) and B cells were also purified into almost homogenous populations from the peritoneal lavage fluid by FACS (Figure 1B) [35].

To obtain in vitro-differentiated MCs, cells from the bone marrow of BALB/c mice were grown in the presence of 30% conditioned media from WEHI-3B cells, which contains IL-3, and with a supplement of 10 ng/mL of recombinant IL-3 for 4 weeks. The resulting cell culture consisted of almost 100% pure MC-like cells, i.e., representing BMMCs. Almost all cells stained positively for Alcian blue and Toluidine blue, although not as strongly as the peritoneal MCs (Figure 1C).

Total RNA was prepared from these cell preparations and subjected to transcriptome analyses by Mouse Ampliseq transcriptome analysis platform, based on the purification of cDNA copies of individual mRNAs on a chip, followed by PCR amplification and individual sequencing. As the RNA is not fragmented, generally every mRNA is therefore read only once and the number of reads will thereby directly match the expression.

Peritoneal MCs were purified from 25 mice, to obtain a sufficient quantity of RNA to allow reliable quantitative analysis of all the 21,000 genes of the mouse genome. The transcriptome analysis presented in this communication has been validated by three independent technologies to ensure correct and quantitative results. As described in a previous article on the transcriptome of mouse MCs, we have used both RNA-seq and Ampliseq, to validate the measurements from several different tissues including BMMCs (5). We have also the possibility to compare these data with an earlier analysis using an unamplified cDNA library [9]. The results have thereby been validated by the combined information from three independent technologies. As we also show in the previous article, the reproducibility with each method is very high (5). It is also noteworthy that previous studies using Northern and Western blot analysis are in agreement with findings in this study [6,10,36,37]. To provide a comprehensive overview of the difference in phenotype between BMMCs and mature peritoneal MCs we selected 153 different transcripts that we could identify as being of particular interest for showing similarities and differences between these two cell populations. The aim has thereby been to give a detailed quantitative view of the similarities and differences between the most commonly used in vitro system for studies of MC biology, namely IL-3 driven mouse BMMCs and peritoneal MCs, to serve as a base for future studies of the in vivo role of MCs in normal immunity and during various diseases.

### 3.2. A Comparative Analysis of Transcript Levels in Peritoneal MCs and BMMCs

The data from the transcriptome analyses of the preparation from mouse peritoneal MCs and BMMCs was arranged with a focus on the genes that were highly expressed in peritoneal MCs but almost absent or expressed at much lower levels in BMMCs (Table 1). Table 2 focuses on genes that are highly expressed in both types of MCs compared to macrophages and/or B cells. Table 3 shows a list of transcripts that are high in BMMCs but low in peritoneal MCs, and Table 4 depicts transcripts that are highly upregulated by LPS treatment for 4 h in cultured BMMCs. Table 1, Table 2 and Table 3 also include the results from the analyses of the peritoneal macrophages and the B cells, allowing a comparison with the transcriptome of MCs.

### 3.3. Transcripts That Were High in Peritoneal Mast Cells but Low or Almost Absent in BMMCs

First, we analyzed for transcripts that are higher expressed in mature peritoneal MCs compared to BMMCs. Interestingly, among these we found several transcripts that are considered highly MC-related or MC-specific (Table 1). As previously shown, the major CTMC chymase, Mcpt4, was almost absent in BMMCs but one of the top four transcripts in the peritoneal MCs. The difference in expression was ~500-fold (Table 1) [5]. Although less pronounced, major differences in expression levels were also seen for the other three major granule proteases. The major tryptic enzyme of CTMCs, Mcpt6, was expressed ~20 times lower in BMMCs compared to the peritoneal MCs. The MC-specific serine protease Mcpt5 (Cma1) was expressed at 8 times lower levels in BMMCs compared to peritoneal MCs. In contrast, Cpa3 showed more similar levels of expression in BMMCs and peritoneal MC with only two times lower levels in BMMCs (Table 2).

A number of other transcripts also showed large differences in levels between these two cell populations. Out of these, the most extreme was the neurotrophin receptor Gfra2, which showed 20,000 times higher expression levels in the peritoneal MCs compared to the BMMCs (Table 1). Moreover, the recently identified receptor for numerous positively charged targets including substance P, Mrgprb2, was expressed 111 times higher in the mature peritoneal MCs compared to the BMMCs (Table 1). Additionally, the C2 complement component was expressed 122 times higher, the metalloprotease Adamts9 was 31 times higher, the histidine decarboxylase was 19 times higher, Milr1 (Allergin) was 6 times higher, and the transcription factors Myb, Meis2 and Tarm1 were expressed at 6, 17 and 7 times higher levels, respectively, in the mature peritoneal MCs compared to the in vitro-differentiated MCs (Table 1).

### 3.4. Transcripts That Were High in Both Peritoneal Cells and BMMCs, but Low in Macrophages and/or B Cells

Table 2 depicts transcripts that were highly expressed in both MC populations compared to macrophages and/or B cells. Among these were the transcription factors Gata2, Tal1, Runx3, Mitf and Zfp521, the cell adhesin molecules Cadm1 and Cadm3, the signaling components Rab27a, Rab44, Lat and Lat2. Such transcripts also included the receptors Mrgpra4, CD200r1 and the von Willebrand factor receptor (Gp1ba), Tubulin alpha 8 (Tuba8), decay accelerating factor (DAF), chemokine Ccl2, chemokine receptor Cxcr1, proteases granzyme B (Gzmb), Mcpt8, Prss34 and Mcpt-ps1, prostaglandin D synthase (Hpgds), heparan sulfate 6-O-sulfotransferase 2 (Hst6st), histamine receptor 4 (Hrh4) and an enzyme for mucin synthesis (Galnt6). Interestingly, the endogenous retrovirus Erv3, that is driving the expression of a Cys-His zinc finger protein, was one of the transcripts in this category [38]. Altogether, these findings reveal that there were numerous transcripts that were high in both BMMCs and mature peritoneal MCs, but low in other hematopoietic cells like macrophages and/or B cells.

We also looked at a few transcripts that were very high in both cell populations and low in macrophages and/or B cells, but that still differed in the levels between the two MC populations. To this category we have two components of the high affinity IgE receptor, FcεRI, the α and the β chains, the transcription factors Gata1 and Runx3, and the bone morphogenic protein 7 (Bmp7) (Table 2).

### 3.5. Transcripts That Were High in BMMCs but Low or Almost Absent in Peritoneal MCs

Table 3 lists transcripts that were present at high levels in BMMCs but were low or almost absent in peritoneal MCs. These included a relatively large number of transcripts, such as the common beta chain of the receptor for IL-3 and GM-CSF (Csfr2b), the cytotoxic T cell ligand A2 (Ctla2a), the thrombin receptor (F2r), and the cytokine receptor signaling component (Clnk or MIST). Among these were also transcripts coding for several cytokine receptors/cytokines including the IL-10 receptor alpha chain (Il10ra), the IL-4 receptor alpha chain (Il4ra) and the IL-2 receptor alpha chain (Il2ra), the G protein coupled receptor Lpar6, the protease inhibitor Serpin3g, CD200r4, CD300lf, the TGF-beta receptor 1 (Tgfbr1), the signaling component Rab38, the prostaglandin E2 receptor Ptger4, the progesterone receptor Pgr, IL-1 beta (Il1b) and IL-6 (IL-6). This category also included the leukemia inhibitor factor (LIF). Notably, LIF was highly expressed in BMMCs but almost absent in peritoneal MCs, macrophages and B cells (Table 3). LIF is crucial for maintaining hematopoietic stem cells in an undifferentiated state and the high levels of LIF in BMMCs may therefore be required for maintaining the immature phenotype of these cells [39,40].

### 3.6. Transcripts That Were Highly Upregulated in BMMCs after Induction by Lipopolysaccharide

A number of transcripts that were highly upregulated after stimulation with *E. coli* lipopolysaccharide (LPS) for 4 h are listed in Table 4. Among these we found several transcripts coding for inflammatory cytokines such as IL-1, IL-6 and IL-13, as well as the TNF receptor Tnfrsf9. We also found an extensive upregulation of transcripts coding for the negative regulator of MC activation: Milr1 (Allergin), the histidine decarboxylase (Hdc), granzyme B (Gzmb), the calcitonin related peptide (Calca), the signal transducer Jak2, and the NFkB inhibitor (Nfkbiz) (Table 4).

The response to LPS by the BMMCs is most likely dependent on the Toll-like receptor-4 (TLR-4). TLR-4, that is activated by LPS, was found at relatively high levels on BMMCs compared to the peritoneal MCs, with 134 reads for BMMCs and 61 for peritoneal MCs (Table 2).

## 4. Discussion

In vitro models of mammalian cells are very important tools for studies of various biological processes. However, relatively few such models are presently available, since only few cell types can grow in vitro without the cell-cell contacts and the growth factors that are supplied by neighboring cells of the organ from where they originate. One example is liver cells: Relatively pure populations of hepatocytes, liver macrophages (Kupffer cells), liver endothelial cells (LECs) and fat cells can be obtained by separating liver cells using collagenase digestion and perfusion, followed by cell purification [41,42]. However, these cells rapidly lose their differentiated phenotype and undergo cell death in culture. There are a few exceptions to this problem. The best example is fibroblasts, which can grow in vitro for months to years and maintain many of their original characteristics [43]. Also T cells can be kept in culture as relatively normal cells, by antigen stimulation [44]. MCs represent a third example, where cultures from mouse bone marrow under the influence of IL-3 and/or SCF can generate essentially pure populations of MC-like cells [31]. BMMCs, phenotypically representing immature CTMCs under standard culture conditions, can be induced to obtain a MMC phenotype by addition TGF-beta or IL-9. In this process, the BMMCs rapidly turn on the expression of the MMC-specific proteases Mcpt1 and Mcpt2 [45,46]. BMMCs have also been instrumental for our understanding of many additional aspects of MC biology. In the literature there is still a misconception that BMMCs has a phenotype of mucosal MCs. However, as mentioned above, BMMCs developed in the presence of IL-3 or IL3/SCF show no similarity to mucosal MCs but instead have a phenotype of immature CTMCs. In support of this notion, they express high levels of the CTMC-specific proteases CPA3, Mcpt5 and Mcpt6, but almost undetectable levels of the mucosal MC specific proteases Mcpt1 and Mcpt2.

However, an important question is how relevant BMMCs are as representatives of tissue MCs? To address this question, we here present a detailed analysis of similarities and differences in the transcriptome phenotype of BMMCs and purified tissue MCs. We have focused on the most commonly used variant of BMMCs, namely cells developed from bone marrow cells in the presence of IL-3. We found that BMMCs had many characteristics in common with peritoneal MCs. However, they also differed in many important aspects. A number of transcripts, including several of the MC-specific proteases, (Mcpt4, Mcpt6 and Mcpt5/Cma1) were expressed at markedly lower levels in BMMCs compared to the peritoneal MCs (Table 1). Moreover, Mrgprb2 (the receptor for substance P and other positively charged low molecular weight compounds), was expressed more than 100 times lower in BMMCs compared to the peritoneal MCs. However, the most extreme case was the neurotrophin receptor, Gfra2, which was almost absent in BMMCs but was expressed at very high levels in the peritoneal MCs. These proteins, together with Adamts9, C2, Milr1 (Allergin), Myb and Tarm1 are most likely examples of genes that are induced late in MC development, and therefore may represent markers for fully mature MCs. Such markers can represent important tools for a detailed analysis of processes that are of importance for the final steps in MC development. Interestingly, Myb was the most highly expressed of all transcriptional regulators in mature MCs, even slightly higher than Gata2 (Table 1 and Table 2) [5]. In contrast, Myb was totally absent in macrophages and was expressed at very low levels in the peritoneal B cells, indicating a prominent role for Myb in MC development (Table 1). It was also interesting to note the relatively low levels of Gata1 in peritoneal MCs compared to BMMCs, indicating that Gata1 is downregulated upon differentiation into fully mature MCs (Table 2). The high levels of leukemia inhibitor factor (LIF) in BMMCs was also of interest (Table 3), and we may speculate that LIF may be one of the factors maintaining BMMCs in an immature state, in analogy with the effect of LIF on hematopoietic stem cells (Table 3) [39,40]. The high basal levels of several inflammatory cytokines in BMMCs compared to peritoneal MCs highlights that caution should be taken when interpreting the physiological relevance of cytokine induction in BMMCs and MC cell lines, as shown in multiple published studies [47,48].

Many additional transcripts were also highly expressed in BMMCs but low or sometimes almost totally absent in mature MCs. Among the most interesting of these transcripts were the common β chain of the receptor for IL-3 and GM-CSF (Csfr2b), cytotoxic T cell ligand A2 (Ctla2a), thrombin receptor (F2r), and the cytokine receptor signaling component (Clnk or MIST). Here we also found several cytokine receptors/cytokines including the IL-10 receptor α chain (Il10ra), IL-4 receptor α chain (Il4ra) and IL-2 receptor α chain (Il2ra), G protein coupled receptor Lpar6, protease inhibitor Serpin3g, CD200r4, CD300lf, TGF-beta receptor 1 (Tgfbr1), signaling component Rab38, prostaglandin E2 receptor Ptger4, progesterone receptor Pgr, IL-1 β (Il1b), IL-6 (IL-6) and LIF. Most of these transcripts were also low or almost absent in peritoneal macrophages and B cells, indicating that they may be characteristic of immature cells and possibly down regulated upon differentiation in many cell lineages.

Basophils and MCs are often considered functional equivalents. However, at least in humans, they represent clearly separate lineages during hematopoietic development [49]. Basophil-like cells can be generated by culturing human bone marrow or umbilical cord cells in the presence of recombinant IL-3 [50]. In these cultures, basophil-like cells appear transiently during the first weeks of culture, and then appear to terminally differentiate and eventually disappear [50]. A phenotypic characterization of these cells (which stain positively with Alcian blue and express FcεRI), showed unexpectedly that their granule content was more eosinophil-like [51]. They expressed high levels of the major basic protein (MBP), the eosinophil cationic proteins (ECP and EDN) and also eosinophil peroxidase (EPO) [51]. This indicates that these in vitro-differentiated cells are hybrid basophilic/eosinophilic cells, in agreement with the close connection between basophils and eosinophils during human hematopoiesis [49]. In analogy with the phenotype of BMMCs compared to peritoneal MCs, this is an example illustrating that in vitro-differentiated cells do not always reflect the phenotype of the tissue-derived corresponding cell population.

In the absence of relevant in vitro models of normal tissue cells, cell lines have become a widely used alternative. A number of cell lines also exist for many cell types. Within the area of hematopoiesis there are immortalized cell lines available for lymphoid cells at many stages of differentiation. There are pre-B-cell lines, B-cell lines and cell lines representing the terminal stage of differentiation of B cells, plasma cells, so called myelomas. Numerous T-cell lines are also available.

In contrast, comparably lower number of cell lines for studies of myeloid cells are available. There are a number monocyte-macrophage cell lines available for studies of both mouse and human hematopoiesis, as well as MC and basophil cell lines [36,52,53,54,55,56]. However, only few cell lines for eosinophil and neutrophil development are available. A widespread cell line used for studies of neutrophil development is HL-60, representing an early myelo-monocytic cell line based on its transcriptome [57]. HL-60 express many of the neutrophil granule proteins, and are thereby one of the few sources of mRNA for these proteins, considering that fully differentiated neutrophils are almost totally devoid of such transcripts due to transcriptional downregulation [57]. HL-60 has been found to express mRNA for a number of neutrophil granule proteins including cathepsin G, N-elastase, MPO, defensin and lactoferrin [54]. For studies of eosinophils, the human cell line Eol-1 is available, expressing low levels of several of the eosinophil proteins [58].

For studies of MCs the human cell lines HMC-1 and LAD2 can be used, which both represent relatively immature MCs. HMC-1 has been shown to express high levels of mRNA for the beta-tryptase and CLC, the Charcot Leyden crystal protein, but almost no chymase, no Cpa3, no alpha-tryptase and very low or undetectable levels of FcεRI alpha and beta chains [53]. The LAD2 and LADR cell lines are likely more differentiated than HMC-1 cells and probably a better alternative, but grow slower [59]. In a more recent study of HMC-1 and LAD2 where a comparison of their phenotype with that of human skin MCs was performed, tryptase levels were found to be comparably low in HMC-1. Further, LAD2 cells were found to express similar IgE receptor levels as normal skin MCs [60]. These studies also showed that both cell lines are relatively poor surrogates of tissue MCs [53,60]. The difference between the two studies may also indicate that the phenotype of these cell lines changes during long term culturing, pointing to the need for careful analysis of the phenotype of a particular sub-line of the cells before using them in an experimental setting. For studies of mouse MCs, a number of cell lines are available which all represent relatively immature MCs, primarily expressing Cpa3 and Mcpt5 but very low level of Mcpt4, i.e., similar to the phenotype of BMMCs [36]. However, several interesting findings have been derived from these cell lines. For example, the IC-2 cell line shows expression of both the early markers of MC development, Mcpt5 and Cpa3, but also the basophil specific protease Mcpt8, indicating that this cell line is arrested in an early stage of MC-basophil development, i.e., before these have separated into different lineages [10,36,61]. This finding together with additional data from cell fate tracing studies during mouse hematopoiesis, have indicated important differences between the mouse and human hematopoiesis concerning the origin of basophils [49,62,63,64]. In mice, MCs and basophils appear to share a common ancestor whereas in human hematopoiesis, basophils and eosinophils are more closely related [49,62,63,64]. For studies of human basophils there are two cell lines showing characteristics of basophil-like cells—KU812 and LAMA-84 [54,55]. KU812 cells express low levels of mRNA for both tryptase and Cpa3, but high levels of mRNA coding for the FcεRI α and γ chains [54,56]. However, KU812 cells also express the erythrocyte proteins α-globin and glycophorin, indicating an immature phenotype [54]. KU812 was also negative for all known neutrophil granule proteins, including cathepsin G, N-elastase, MPO, defensin and lactoferrin [54]. LAMA-84 expressed only very low levels of the FcεRI alpha chain and almost no Cpa3, but instead expressed high levels of CLC [55].

As indicated above, cell lines can in many cases be relevant alternatives for studies of for example, cell signaling, receptor function, cell adhesion and cloning. However, it is important to keep in mind that cell lines are often arrested in early, proliferative stages of hematopoietic cell development, and that they, due to long term growth in vitro, have many chromosomal abnormalities and thereby can be relatively poor alternatives to in vivo cells and whole animal studies. Similarly, BMMCs have their limitations. For example, several studies have indicated a crucial role for the signal transducer of cytokine signaling Clnk (MIST) in IgE receptor signaling in MCs using BMMCs as a model [65,66,67,68,69]. However, in this study we found that MIST expression was essentially absent is in mature peritoneal MCs. Hence, MIST has probably only a minor role in tissue MCs. Highly divergent results between BMMCs and peritoneal MCs have also been obtained when comparing the effect of Mrgprb2 triggering on Ca^2+^ mobilization, with peritoneal MCs responding robustly whereas varying responses were seen in BMMCs [70]. Most likely, this is an effect of the differential expression of Mrgprb2 seen in this study, where we found high Mrgprb2 expression in peritoneal MCs but only minute expression in BMMCs (Table 1). Recently, both Gata1 and Gata2 have been shown to be important for the expression of Mcpt6 in mouse BMMCs [71]. However, Gata1 showed very low expression in peritoneal MCs compared to BMMCs, indicating that Gata1 most likely has only a minor role in the regulation of Mcpt6 transcription in mature MCs compared to BMMCs (Table 2). Notably, it has been shown that Gata2 has an essential role in MC development in zebrafish, whereas Gata1 does not appear to influence MC development in this species [72].

As shown by this and several previous studies, mouse BMMCs developed in the presence of IL-3 show a relatively immature phenotype [5,73]. The question is then if alternative protocols for obtaining BMMCs can be used to obtain cells with a more mature phenotype. Using a combination of IL-3 and SCF, or SCF alone, seems to give a slightly more mature phenotype. An analysis of the transcriptome of BMMCs cultured with IL-3 + SCF has been performed, primarily focusing on seven MC- and basophil-related granule proteases. These cells were found to express a very similar protease profile as seen in this study. However, a higher expression level of the MC-specific tryptase, Mcpt6 was observed [71]. An alternative to using recombinant cytokines for the development of BMMCs is to use fibroblasts as feeder cells. An analysis of cocultures of IL-3-driven BMMCs with fibroblasts was shown to result in a potent upregulation of several important MC-expressed genes including a ~60-fold upregulation of Mcpt4, a 10-fold upregulation of histidine decarboxylase (Hdc) and a 5-fold upregulation of the endothelin receptor A (Ednra) [74]. The fibroblasts used in the latter study were found to express primarily the membrane-bound form of SCF [74]. The very potent upregulation of Mcpt4 by the co-culture with fibroblasts gives a strong indication that other factors than the fibroblast SCF are of importance for induction of Mcpt4, since the levels of Mcpt4 expression in BMMCs were only marginally increased by culture with SCF [71]. However, even in the presence of fibroblasts, the Mcpt4 expression levels are only ~10% of the levels seen in vivo, indicating that the local tissue environment supplies other factors of major importance for the development of fully mature MCs.

Mouse BMMCs developed in the presence of IL-3 show a relatively immature phenotype compared to peritoneal MCs. An obvious question is then if MCs from other tissues are more similar to these BMMCs. In the lungs, the number of MCs and basophils are very low and the phenotype of these cells appear to be very heterogenous (5). In our previous studies, we found very low transcript levels for all of the MC- and basophil-related proteases (5), the most abundant transcript being Mcpt5 (only 8 reads). The Mcpt4 and Cpa3 transcript levels were also very low (3 and 1 reads, respectively) (5). The low levels of Cpa3 in the lung is in marked contrast to the situation in BMMCs, where Cpa3 is the dominating protease transcript (Table 2) (5). To put these expression levels in a context, Mcpt5 gave 45,221 reads in peritoneal MCs (corresponding to a ~5700-fold increase vs. lung), emphasizing the low number of MCs populating the naïve mouse lung (5). Low numbers of MCs in the lung of naïve mice is confirmed in other studies [75]. In the latter study, the few MCs found in the lung were located primarily around and/or between larger airways and blood vessels [75].

In our previous study, in most other tissues except duodenum and the spleen, the dominating MC-specific protease transcripts were Mcpt5, CPA3, Mcpt6 and Mcpt4 (5). All of these transcripts are characteristic of CTMCs. The uterus, the tongue and the peritoneal MCs showed very similar relative levels of these four proteases. The tissue that differed most extensively concerning the levels of these transcripts was the ear, where Mcpt4 was twice as abundant as the other three proteases (5). Notably, single cell analysis has given a slightly different picture [76]. In the latter study, it was indicated that the expression levels for a majority of transcripts were very similar between different MC subpopulations but there was also a number of transcripts that differed between the tissues. Also in that study, the population differing to the highest extent was the skin MCs, where the transcript levels for Adamts1, Adamts5, Gfra2, Mrgprb13 and 8 were considerably higher in skin MCs vs. other MC subpopulations [76]. Notably, human MCs originating from different regions of the skin may also differ in phenotype [77], and there is also a reported large variation in the expression levels for a number of MC-specific transcripts between individuals [78]. Altogether, we can therefore conclude that there are clear differences in MC phenotype between different tissues, between individuals and between different regions of the same tissue. The majority of tissues have variations in the CTMC population, whereas the duodenum shows almost exclusively a population of T-cell dependent MMCs. Further, mouse lung and spleen have very low levels of MCs and basophils, and with a high degree of heterogeneity within these populations.

Importantly, none of the MC populations found in vivo in mice show a phenotype that clearly resembles that of BMMCs, and it is therefore unlikely that MCs in any tissue have a similar phenotype as BMMCs. One interesting possibility is thus that BMMCs instead are more similar to MC precursors coming from the bone marrow. Such precursors can under inflammatory conditions enter inflamed tissues. After entering the tissue, these MC precursors most likely obtain a more mature phenotype by coming in contact with the local tissue cells and factors produced by the tissue. A single cell analysis of these MC precursors would most likely give more conclusive information concerning this possibility.

That MCs also can change phenotype markedly as a response to various stimuli and upon various pathological conditions has been shown in several model systems [79]. The MC phenotype can change upon bacterial infection, by TLR-signaling (as shown in Table 4), by UV-exposure, by increasing extracellular ATP concentrations after injury, in tumors and in the skin and lung under allergic conditions, as summarized recently [79]. This highlights the complexity of analyzing the in vivo function of MCs as the phenotypes of these cells can vary extensively between tissues and also under various inflammatory conditions. Thereby, it may be argued that in vitro models can be relatively poor models for studies of the in vivo function of MCs.

In conclusion, this study highlights the importance of having detailed knowledge of the phenotype of model cell population in use, in order to evaluate the functional impact of various molecules in a particular cell type. This study, where similarities and differences in the transcriptome between the immature BMMCs and mature peritoneal MCs were defined, can thereby serve as such a guide to assess which experiments that are relevant to be performed in BMMCs compared to more mature MC populations. Such information is of crucial importance for efforts aimed at elucidating the biological function of MCs.

## Figures and Tables

**Figure 1 cells-09-02118-f001:**
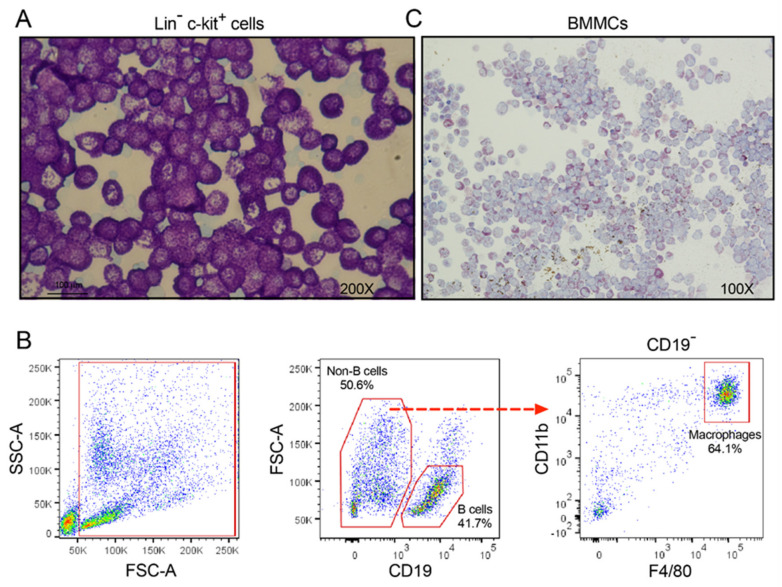
Freshly isolated peritoneal mast cells, B cells and macrophages as well as in vitro-differentiated mast cells. (**A**) A representative image of freshly sorted peritoneal mast cells. Through MACS negative selection, lineage negative (CD3^−^, CD4^−^, CD8^−^, CD11b^−^, CD19^−^, B220^−^, Gr-1^−^ and TER-119^−^) peritoneal cells were obtained and subsequently subjected to positive selection using MACS c-kit Micro Beads. The obtained Lin^−^ c-kit^+^ cells gave rise to more than 95% pure toluidine blue-positive mast cells. (**B**) Representative gating strategy used for sorting of peritoneal B (CD19^+^) cells and macrophages (CD19^−^ CD11b^hi^ F4/80^hi^) by flow cytometry. After excluding debris in the first gate (FSC-A vs. SSC-A), doublet cells were also excluded using FSC-A vs. FSC-H and SSC-A vs. SSC-H plots (not shown). (**C**) A representative image of toluidine blue-stained bone marrow-derived mast cells (BMMCs) after growing in the presence of WEHI-3B-conditoned medium and recombinant IL-3 for four weeks.

**Table 1 cells-09-02118-t001:** Transcripts found at high levels in mature peritoneal MCs but at low levels in BMMCs. The table depicts the number of reads for each transcript. The right column depicts the fold difference in transcript levels between P-MCs vs. BMMCs.

Transcript	BMMCs	P-MCs	MQ	B Cells	Fold Dif
Mcpt4 (mMCP-4)	7	31,290	5	10	(500×)
Tpsb2 (Mcpt6)	3119	67,773	6	17	(20×)
Cma1 (Mcpt5)	5683	45,221	4	11	(8×)
Mrgprb2 (Substance P rec.)	8	899	0.1	0.2	(111×)
Gfra2 (Neurotrophin rec.)	0.1	2016	7	1	(20,000×)
Adamts9 (Protease)	12	370	1	0	(31×)
C2 (Complement f. 2)	1.3	159	5	2	(122×)
Milr1 (Allergin)	48	283	24	23	(6×)
Hdc (Histidine decabo×ylase)	48	936	42	2	(19×)
CtsE (Cathepsin E)	336	1248	75	336	(4×)
Serpinb1a (Protease inh.)	460	1037	20	64	(2×)
Myb (Transcr. f.)	392	2490	0	9	(6×)
Meis2 (Transcr. f.)	18	300	0	0	(17×)
Tarm1 (Transcr. f.)	21	156	0	0	(7×)
Il3ra (IL-3 rec. alpha)	25	159	17	4	(6×)

BMMCs, bone marrow-derived MCs; P-MCs, peritoneal MCs; MQ, macrophages. rec., receptor; inh., inhibitor; f. factor.

**Table 2 cells-09-02118-t002:** Transcripts found at high levels in both BMMCs and mature peritoneal MCs but at low levels in MQ and/or B cells. The table depicts the number of reads for each transcript. The right column depicts fold-differences in transcript levels between P-MCs vs. BMMCs for a few selected transcripts.

Transcript	BMMCs	P-MCs	MQ	B Cells
Cpa3 (CPA-3)	22,478	45,604	1	6
Gata2 (Transcr. factor)	5205	2272	0	1
Fcer1a (IgE rec. alpha)	1631	345	0	0 (5× lower in P-MCs)
Ms4a2 (IgE rec. beta)	4288	1297	0	2 (3× lower in P-MCs)
Fcgr3 (IgG rec. III)	538	752	1968	27
Cadm1 (Cell adhesion)	913	1118	4	9
Rab27b (GTP-ase)	856	858	0	0.7
Inpp5d (Phospatase)	826	663	124	287
Cited2 (Trans act.)	772	599	17	30
Tmem9 (Trans memb. pr.)	767	456	52	25
Basp1 (Brain acid pr.)	686	1005	6	2
CD200r3 (Surface rec.)	640	275	1	0.1
Tuba8 (Tubulin alpha 8)	622	439	0.2	0.6
Maob (Mono amine ox.)	577	359	2	0.1
Erv3 (End. retrovirus)	514	141	0.2	0.7
CD55 (DAF)	510	439	12	248
Rgs18 (Reg. of G prot.)	487	509	79	10
Ccl2 (Chemokine)	427	397	1.2	0.4
Pik3r6 (PI3k Subunit)	423	767	179	9
Slc45a3 (Solute carrier)	419	673	0.8	0.2
CstF (Cystatin F)	388	181	0	0.1
GzmB (Granzyme B)	386	581	0	0
Dock10 (Dedic. cytok.)	383	389	102	332
Bmp7 (Bone morf. pr. 7)	375	56	0.2	0 (7× lower in P-MCs)
Slc30a2 (Zinc transp.)	369	220	0	0
A4galt (enz. galact. cer)	364	71	0	0
Specc1 (Cytospin-B )	300	124	14	18
Cadm3 (Cell adhesion)	298	118	0.4	0.1
Gata1 (Transcr. f.)	296	74	0	0 (4× lower in P-MCs)
Samsn1 (Neg. reg. B-cl)	285	187	22	15
Rab44 (Ras fam. memb.)	276	302	1.3	0.1
Dapp1 (Rec. signaling)	270	389	96	71
Grap2 (Cell signaling)	214	261	5	2
Hpgds (Prost.gl. D synt.)	233	450	2	2
Gm973 (Predicted gene)	232	261	0	0.4
Slc2a3 (Glucose transp.)	208	102	0.5	41
Slc6a12 (Solute carrier?)	205	191	0	0.7
Nrn (Nurin nuclear env.)	204	208	22	145
Smpx (Small muscle pr.)	203	233	0	0.1
Nfe2 (Transcr. f.)	188	108	126	3
Abcb1b (ATP dep. trans.)	189	336	37	6
Blm (Bloom syndr.)	175	73	3	10
Mrgpra4	163	155	0	9
Tal1 (helix-l-h transc.f.)	160	246	3	0.1
CD200r1 (Cell surf. rec.)	150	246	3	0.1
Rnf180 (Ring f. Ubiq. lig.)	150	156	5	0
Runx3 (Transcr. f.)	143	409	7	65 (3× lower in BMMCs)
Mitf (Transcr. f.)	144	370	36	3
Tlr4 (TLR-4)	134	61	200	26
Rgs1 (Reg. G-prot. sign.)	132	51	1	0
Lat (Signaling)	130	111	0.4	1.2
Lat2 (Signaling)	2053	1588	291	546
P2rx1 (ATP rec)	129	327	69	2
Rgs13 (Reg. G-prot. sign.)	126	289	0.2	0.5
Dgki (Diacyl glyc. kin.)	125	33	0	0
Galnt6 (Mucin synth.)	123	122	1	31
Bcl2 (Anti apoptotic)	132	101	10	15
CD69 (Leukocyte ag.)	110	45	0.5	123
Gpr141 (G-prot. c. rec.)	109	47	0.3	0
Gp1ba (vWF-receptor)	93	61	0	1
Tespa1 (thymoc. exp.)	85	153	0	17
Gcsam (Germinal c. ass.)	82	311	0	0.4
Spn (CD43 Leukosialin)	72	134	0	5
Cx3cr1 (Fractakine rec.)	62	28	0	0.2
Ryr3 (Ryanodine rec.)	59	24	0	0.3
Atp8a2 (Phopholipid tr.)	59	41	0.5	0.1
Gsg11 (Germ cell ag.)	58	27	0	0.4
Mcpt8 (Basophil Prot.)	46	38	0.1	0.1
Hst6st (Heparan s.-O-S)	44	152	0	0
Kcne3 (Potassium ch.)	42	112	2	0.4
Prss34 (Mcpt11)	41	257	0	5
Fcgr2b (Fc-γ rec. 2B)	40	91	9	72
Mcpt-ps1	36	210	0	0
Zfp521 (Zinc finger g.)	29	67	0	0
Hrh4 (Histamin rec.4)	26	25	0	0
Il18r1 (IL-18 rec. alpha)	26	24	0.1	2
Il9r (IL-9 receptor)	25	11	0	73
Gfi1 (Zinc finger prot.)	21	79	0.1	0.8
Dfna1 (B lymfo blasts)	20	54	1	0
Il4 (IL-4)	14	13	0	0
Il13 (IL-13)	11	4	0	0

BMMCs, bone marrow-derived MCs; P-MCs, peritoneal MCs; MQ, macrophages. Rec., receptor; inh., inhibitor; f. factor; ch, channel; g., gene; ag., antigen; tr., transporter; ass., associated; kin., kinase; dep., dependent; End., Endogenous; synth., synthase;

**Table 3 cells-09-02118-t003:** Transcripts found at high levels in BMMCs but at low levels in mature peritoneal MCs. The table depicts the number of reads for each transcript. The right column depicts the fold difference in transcript levels between BMMCs vs. P-MCs.

Transcript	BMMCs	P-MCs	MQ	B Cells	Fold Dif.
Csf2rb (IL-3+GM-CSF rec. beta)	3485	415	150	84	(8×)
CTLA2a (CTL assoc. prot.)	1510	5	2	2	(302×)
F2r (Thrombin receptor)	1463	12	0	0	(122×)
F13a1 (Coagulation factor 13)	994	6	122	5	(166×)
Clnk (MIST, cell signaling)	921	4	0.1	0	(230×)
IL10ra (IL-10 rec. alpha)	919	49	280	268	(19×)
Rnf128 (Ubiquitin ligase)	889	78	28	0.5	(11×)
Lpar6 (G prot. coupled rec.)	825	70	35	105	(12×)
Serpina3g (Protease inhibitor)	710	5	0.3	60	(142×)
Nampt (enzyme)	706	55	69	48	(13×)
Lif (Leukemia Inh. factor)	584	4	0.5	0.1	(146×)
Gab2 (Cell signaling)	583	45	115	17	(13×)
Rnase6 (Anti bacterial)	557	26	7	153	(21×)
Pik3cd (PI3K delta)	507	64	102	319	(8×)
CD200r4 (Cell surf. rec.)	487	21	37	0.6	(23×)
Sema4d (CD100)	480	47	3	132	(10×)
Anpep (Alanine-amino pep.)	447	118	19	2	(4×)
Dnm3 (Dynamin)	435	98	0	0	(4×)
Dtx4 (Ubiquitin Ligase)	384	7	18	14	(55×)
Rab38 (Ras rel. prot.)	375	6	0.3	0	(62×)
Akr1c12 (Alpha-keto red.)	366	2	0.8	0.2	(183×)
Aqp9 (Aquaporin)	360	2	46	0.5	(180×)
Spns3 (Sphingolipid transp.)	346	12	0	15	(29×)
Tgfbr1 (TGF beta receptor)	319	96	36	27	(3×)
Krba1 (KRAB-A cont.)	304	34	7	57	(9×)
Stap1 (Sign.trans B cells)	300	14	1	150	(21×)
Stk19 (Ser/Thr kinase)	299	32	48	52	(9×)
Glipr1 (Cys rich secr. pr.)	292	29	5	20	(10×)
IL-4ra (IL-4 receptor alpha)	282	12	21	23	(24×)
Treml2 (Tr. rec. myeloid)	278	4	0.3	72	(70×)
Avil (Advillin)	270	0.7	0	0.5	(386×)
Tmem233 (Trans memb. pr.)	269	2	0	0.1	(134×)
Mthfd2 (Mitoch. enzyme)	266	8	6	20	(33×)
Tbxas1 (Thrombox. synthase)	243	23	103	2	(10×)
CD300lf (Membr Glycopr. myeloid)	239	49	10	63	(5×)
Birc5 (anti-apotosis)	234	41	13	4	(6×)
Neb (Nebulin actin binding.)	211	0	1	0.5	(≈211×)
Kcnn4 (Potassium channel)	202	10	3	63	(20×)
Il2ra (IL-2 rec. alpha)	196	14	0.2	8	(14×)
Nrop3 (Nucl. rec. interacting)	165	2	0	0	(82×)
Ptger4 (Prostagl. E2 receptor)	165	11	116	24	(15×)
Pgr (Progesterone receptor)	147	5	0	0	(29×)
Gpr174 (G-prot coup rec?)	145	24	1	137	(6×)
Nlrp3 (NALP3 Inflammasome)	134	8	90	1.4	(17×)
Ahrr (Aryl hydr. carb. receptor)	134	1.2	7	0.1	(112×)
Calca (Calcitonin related pept.)	133	1.2	0	0	(111×)
CD96 (Tactile T-cell act.)	121	3	0.5	0.5	(40×)
Gfi1b (Zinc finger)	106	25	11	3	(4×)
Il1b (IL-1 beta)	109	4	4	0.4	(27×)
Greb1 (Estrogen resp. gene)	107	1.7	0	0.3	(63×)
Ulbp1 (Stress ind. NK rec.)	96	3	2	7	(32×)
Rhof (Ras hom. fam.)	95	10	0.3	251	(10×)
Cited (Trans activator)	94	3	0.3	1.4	(31×)
Cyp4a12a (Cytochr. P450)	86	0	0	0	(≈86×)
Aqp8 (Aqua porin)	64	0	0	0.1	(≈64×)
Il6 (IL-6)	64	12	1.2	0	(5×)

BMMCs, bone marrow-derived MCs; P-MCs, peritoneal MCs; MQ, macrophages. rec., receptor; pr., protein.

**Table 4 cells-09-02118-t004:** Transcripts that are highly upregulated in BMMCs after LPS stimulation (4 h). The table depicts the number of reads for each transcript. The right column depicts the fold difference in transcript levels between BMMCs + LPS vs. unstimulated BMMCs.

Transcript	BMMCs	BMMCs + LPS	Fold Dif.
MilR1 (Allergin)	48	332	(7×)
Hdc (Histidine decarb.)	48	279	(6×)
Nfkbiz (NFkB inh.)	42	422	(10×)
Mgat5 (Oligosaccharide enzyme)	43	331	(8×)
Tmem63b (Ion channel)	59	397	(7×)
Il1b (IL-1 beta)	109	1804	(17×)
Il6 (IL-6)	64	246	(4×)
Tnfrsf9 (TNF rec. superf. 9, CD137)	8	459	(57×)
GzmB (Granzyme B)	386	1900	(5×)
Gzmc (Granzyme C)	0.1	110	(1100×)
Calca (Calcitonin related peptide)	133	917	(7×)
Il13 (IL-13)	11	83	(8×)

BMMCs, bone marrow-derived MCs.

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
