# Peer review of "How Relevant Are Bone Marrow-Derived Mast Cells (BMMCs) as Models for Tissue Mast Cells? A Comparative Transcriptome Analysis of BMMCs and Peritoneal Mast Cells"

_cells, 2020, doi:10.3390/cells9092118_

Round 1

Reviewer 1 Report

Authors report a transcriptome analysis of peritoneal mast cells (MCs) and bone marrow-derived mast cells (BMMCs) in order to determine whether BMMCs can be considered as an equivalent of tissue-resident MCs. Authors compare transcripts expressed in BMMCs with those expressed in peritoneal MCs in basal conditions and find important differences in the expression of several genes. Finally, authors stimulate BMMCs with LPS and analyze their transcriptome. Although some novel findings are presented, several important issues have to be addressed.

  1. Although this investigation contributes to the knowledge about the differences in expressed genes by different MC preparations, the relevance of the results is limited since the notion that BMMCs cannot be considered equivalent to tissue-resident mast cells has been largely present in the field and the extreme plasticity of MC transcriptome has been documented by others and by this group of research (Frossi, B., et al., Immunological Reviews, 2018; Dwyer,D., et al., Nature Immunology, 2016; Akula, S., et al, Cells, 2020). Also, even differences in BMMCs obtained from different strains of mice has been reported (Yamashita, Y. et al. J Immunol, 2007). Whit this in mind, it is necessary to justify the comparison between this specific preparation of Balb c-mice-derived BMMCs (grown in the presence of the supernatant of WEHI cells plus IL-3, which will generate a specific transcriptome) with peritoneal MCs.

  1. Authors shown in Figure 1A a picture of freshly isolated peritoneal MCs and this picture is exactly the same that was shown on Figure 1B of a previous paper (Akula, S., et al. Cells 9:211, 2020). Also, the description of the Thermo Fisher mouse Ampliseq transcriptome analysis platform (lines 165 to 168 of the present manuscript) are the same that those presented on the mentioned paper. Authors must remove Figure 1A and refer to previous work and re-phrase the description of the platform.

  1. There is no statistics paragraph in the Material and Methods section. Please explain how data were analyzed. Also please specify the serotype of LPS that was used. 

  1. Tables show Fold Dif, which refers to the fold difference in transcript levels between P-MCs vs BMMCs (lines 240-241). However, it is not clear how those data were obtained from the numbers that appear in the BMMCs and P-MCs columns. Please explain or verify the procedure to obtain Fold Dif. On the other hand, Table 2 is not correctly formatted (please check the number of columns and column heads).

  1. Lines 307 to 309 state that “cultures from mouse bone marrow under the influence of IL-3 and/or SCF can generate essentially pure populations of MC-like cells”. However, it has been demonstrated that cultures of MCs in the presence of IL-3 are similar to mucosal MCs and addition of SCF leads to the generation of connective tissue-MCs (Ito, T., et al., J Immunol, 2013; Nakano, T., et al., J Exp Med, 1985; Tsai, M., et al., PNAs, 1991). Please modify the sentence and cite the mentioned papers.

6. The experiment of treating only BMMCs with LPS is not justified. Authros must explain why peritoneal MCs were not treated with LPS as well and how the presented results with LPS-treated BMMCs strenght the main conclusion of the manuscript.

Author Response

Response to reviewer 1.

BMMCs grown in the presence of IL-3 is the most commonly used model system for in vitro studies of mast cells why a detailed characterization of them should have been done many years ago. Now we have performed such an analysis involving all the 21 000 genes of the mouse genome and studied it by two independent technologies RNA seq and Ampliseq. In this study we do not only describe a few transcripts as in previous studies but give examples of 153 different genes that in some respect are illustrating the phenotype of the BMMC compared to a more mature connective tissue mast cell and to two other populations of immune cells namely peritoneal B cells and peritoneal macrophages, to put them in a broader context. We also give quantitative information which almost never have been given before except in our previous paper in Cells. However, now with a much larger number of transcripts focusing on BMMC phenotype. This is really the first such detailed and also quantitative study performed. The single cell analysis that have been published are informative but poor tools for real quantitative studies.

  1. We have in the previous article Akula et al 2020 shown that the phenotype of mast cells in different tissue differ quite extensively, particularly between skin, lung and duodenum. However, it is few studies if any that show that peritoneal mast cells or mast cells of the skin differed markedly from one experiment to another. There are clearly strain differences. Some strains lack the expression of Mcpt7 but has similar levels of other proteases. The Frossi paper is a nice review of changes in mast cell numbers and marker expression, with the focus on the effect by different stimuli, like bacterial infection, parasite infection and deliberate deletion of specific subpopulations but little if any on difference under identical conditions- so we disagree that mast cells under the same conditions would differ markedly.
  2. We have now exchanged the figure to another never shown before and at a higher magnification. The Materials and methods section has also been rewritten, to say the same thing but with other words.
  3. As now stated in a separate section added to the beginning of the results section this analysis is based on the results from more than three totally independent types of studies, RNA-seq, Ampliseq and an unamplified cDNA library. To this comes supporting data from Northern and Western blot results from several previous articles which gives a much higher degree of reliability than performing the same experiment with the same technology several times. To perform such an analysis of peritoneal cells, would have involved the sacrifice of 100-120 mice. The result would also most likely have been almost the same. We have namely seen in a not published part of the Ampliseq study where 4 samples were from 4 different mice studying the pancreas. The data from these 4 totally independent mice were almost identical showing the reproducibility of the analysis. So we do not think its ethically defendable to perform such a repetition involving 100-120 mice, giving an almost identical result. The LPS was bought from Sigma-Aldrich and from E. coli O127:B8. The catalog number of the LPS has also been added to the Materials and Methods section.
  4. The fold difference is calculated from the number of reads in the two cell types of the particular transcript. The Table is correct in our version on the Mac. However, I have heard that it sometimes becomes rearranged when opening on a PC. Look at the accompanying pdf. There it is probably correct on your computer. The Journal also reformats Tables to their own format, and we can recheck in the proof when it comes (hopefully).
  5. The statement that BMMCs have a phenotype of mucosal mast cells is an old misconception that has become stuck in the literature. This in not correct as we also clearly stated this in the beginning of the discussion. We have now also added a section in the discussion to more clearly state that BMMCs do not show any resemblance with mucosal mast cells but of immature connective tissue mast cells. We hope that this manuscript can help in removing this misconception from the field.
  6. The LPS treatment of the BMMCs were added to give an extra angel on this in vitro model for mast cells and what genes are the major genes that are upregulated by LPS. To do the same experiment with peritoneal cells we would most likely have needed to sacrifice 50-100 more mice and our permit did not allow for such a large number and to get a new permit would have taken us half a year why we decided not to include such a study as this would not to have been possible within the time frame of this special issue of Cells. It is also not the main focus of the manuscript just as I said to give a second angle on these cells and how they respond to an important bacterial component.

Reviewer 2 Report

Akula S et al. expanded on their previous study focused on the analysis of the mouse mast cell transcriptome (PMID 31947690) to investigate whether bone marrow-derived mast cells can serve as a model for tissue mast cells. My main concern here is that the analysis is somehow incomplete as the authors only looked into differences in the gene profile of BMMCs vs. peritoneal mast cells. Is it possible that BMMCs are more similar to mast cells from other tissues other than the peritoneum? From my reading of their previous publication, I understand that the lung mast cell transcriptome was already studied and can be used for comparison.

I also do not understand the relevance of the data shown in Table 4 as it does not address the question posed by the study.

Author Response

Response to reviewer 2.

Lung mast cells are extremely few and very heterogenous as shown in our previous paper. To purify the different subpopulations of mast cells of the lung for a transcriptome analysis of the quality presented in this manuscript we would most likely have need to sacrifice 300-500 mice, if even possible. This because the mast cells of the lung are so few and also stuck in the tissue why extensive collagenase and protease treatment would be needed to get them out of the tissue, which may severely harm the cells why the yield would drop significantly. So it would most likely not be possible. The mast cells of most other tissue are very similar in phenotype to the peritoneal mast cells as we also show in the previous Cells paper except for the mucosal mast cells of the duodenum. However, these mucosal mast cells of the duodenum show very little in common with the IL-3 driven BMMCs why peritoneal MCs clearly is the most feasible and relevant mature mast cells for a comparative analysis with BMMCs.

                      The LPS treatment of the BMMCs were added to give an extra angel on these in vitro model for mast cells and what genes are the major genes that are upregulated by LPS.

Reviewer 3 Report

The work of Akula et al is a reanalysis of their previously published RNA Sequencing dataset (Akula et al Cells 2020 Jan 14;9(1):211. doi: 10.3390/cells9010211.) comparing different types of mast cells. That previously published work is very useful to the community. The data is essentially the same as the earlier paper in this work, although the analysis is more focused on comparing BMMCs and tissue mast cells and extends the utility to the field. However, this work includes sections (in the materials and methods section, e.g.) that is significantly the same as the previous work. In addition, this submission actually includes a figure from that previous paper (Figure 1A in this paper and Figure 1B in the previous paper). The authors should ensure that the overlaps between the 2 papers are removed.

Author Response

Response to reviewer 3.

We have now exchanged the figure to another never shown before and at a higher magnification. The Materials and methods section has also been rewritten, to say the same thing but with other words.

Round 2

Reviewer 1 Report

Comments to authors:

  1. Statistics methods must be shown
  2. References about MC gene expression heterogeneity in tissues must be included
  3. Tables 3 and 4 must be correctly formatted

Author Response

Reviewer 1

Dear Reviewer,

  1. The Peritoneal mast cells are performed on the material from 25 mice to get sufficient material to cover all 21000 mouse genes with a high quantitative estimate of the transcript abundance of all these genes. On the chip we have material from four individual mice where the result is almost identical between these four mice. To perform a similar experiment on the peritoneal cells would involve the sacrifice of 125 mice to get five identical samples and half a year of work, where we expect the result to be almost identical as the pancreas samples, which I do not think is ethically defendable. To ensure an even better reliability of our results we have performed the analysis by two completely independent technology- RNA-seq performed by GATC-biotech in Germany and the ThermoFisher Ampliseq performed here in Uppsala with very similar results as in our previous article in Cells on the mast cell transcriptome. The results are also validated by Northern and Western blots for some of the genes in previous articles as referenced in the manuscript showing a very high reliability of the data. We have also performed a similar analysis on rat peritoneal mast cells in an earlier J.Exp.Med article using plaque screening of an unamplified cDNA library that shows very similar transcript levels as our now mouse data so the data is much better validated than any other study including the extensive single cell studies published in high impact journals. Our study is thereby validated by in total 5 independent studies, which in our minds give a much better validation than performing the same study five times with almost identical values. That is why we have no statistics as you cannot do statistics on five different independent methods but the reliability is superior to any statistical method and any data previously published on peritoneal mast cells and BMMCs.
  2. Almost a full additional page including many new references on mast cell heterogeneity has been included in the updated manuscript marked in red.
  3. I have rechecked the tables and they are correct on my computer and also on the manuscript I downloaded from the Cells home page. However, I have seen that when we send the tables in the manuscript to co-authors that have PCs it sometimes occurs rearrangements. However, as I said everything looks fully correct in the version that I uploaded and also downloaded from the Cells homepage why it probably is a PC phenomenon. I hope the journal have a Mac to see it correct and as we also send the manuscript in a PDF format that can be uploaded by PC users if their computers induce aberrant formatting. The journal also reformats the Tables to their style that we can correct in the Proof if there has been any aberrant reformatting at the journal upon processing the manuscript.

Hope my clarifications are satisfactory.

Reviewer 2 Report

This reviewer understands that it is very difficult to obtain large number of mast cells from lungs or other tissues to compare their transcriptome from that generated by bone marrow-derived mast cells. However, I still think that this study partially answers the question posed in the title as the transcriptome of peritoneal mast cells is known to be different from the transcriptome of mast cells located in other tissues. This needs to be addressed in the discussion.

Also, the LPS data clearly diverts from the goal of the study. The authors should reconsider their inclusion in the manuscript.  

Author Response

Dear Reviewer

  1. Almost a full additional page including many new references on mast cell heterogeneity has been included in the updated manuscript marked in red. In this section we have also addressed the changes in phenotype that occur upon stimulation and the role of TLRs in such change in phenotype, which includes the Table 4. This puts the LPS study in a context of the other studies of the manuscript.

We hope these clarifications and addition is what you requested.

Reviewer 3 Report

The authors have replaced the figure that was previously published.

Author Response

Dear reviewer,

Thanks or your comments.